# Correlation of Biomarkers with Endoscopic Score: Ulcerative Colitis Endoscopic Index of Severity (UCEIS) in Patients with Ulcerative Colitis in Remission

**DOI:** 10.3390/medicina57010031

**Published:** 2020-12-31

**Authors:** Corina Silvia Pop, Petruta Violeta Filip, Sorina Laura Diaconu, Clara Matei, Florentina Furtunescu

**Affiliations:** 1Carol Davila University of Medicine and Pharmacy, 020021 Bucharest, Romania; cora.pop@gmail.com (C.S.P.); sorinadiac@yahoo.com (S.L.D.); matei_clara@yahoo.com (C.M.); florentina.furtunescu@umfcd.ro (F.F.); 2Department of Internal Medicine and Gastroenterology, University Emergency Hospital of Bucharest, 050098 Bucharest, Romania; 3Colentina University Hospital, 020125 Bucharest, Romania

**Keywords:** ulcerative colitis, fecal calprotectin, C-reactive protein, hemoglobin concentration, endoscopic remission

## Abstract

*Background and Objectives:* Ulcerative colitis is a disease with an unpredictable evolution, often highlighted endoscopically, that is associated with persistent inflammation affecting the patient’s quality of life. An attempt was made to discover surrogate markers to evaluate the endoscopic remission of the disease in order to increase the patient’s quality of life and also their adherence to the treatment and monitoring plan. One such marker is fecal calprotectin (FC). To confirm the correlation between biomarkers and endoscopic disease activity and to define the optimal cut off value to detect clinical and endoscopic remission in a center of Romania. *Materials and Methods: * This was a prospective study that included 59 patients diagnosed with ulcerative colitis at the Department of Internal Medicine III, University Emergency Hospital of Bucharest. Patients had fecal calprotectin measurements and colonoscopy/rectosigmoidoscopy performed during baseline, 6 and 12 months. For endoscopic activity the Ulcerative Colitis Endoscopic Index of Severity (UCEIS) was used. *Results:* During the study, relapses have occurred in 35.6% of patients, the median age was 47 years (21–77). During the study, the FC measurement was significantly increased at 3 months (median, range µg/g; 715, 14–4000) and at 6 months (median, range µg/g; 650, 4.5–3000) (*p* ≤ 0.05). Another inflammatory biomarker studied was CRP, which showed increased values at 3 months (median, range, mg/dL; 1.86, 0.14–58.9), at 6 months (median, range, mg/dL; 2.36, 0.12–45.8) and at 9 months (median, range, mg/dL; 2, 0.12–25.9) compared to the baseline (*p* = 0.01). Patients with recurrence of the disease also associated an increase in the values of clinical evaluation scores (SCCAI; *p* = 0.00001), but also endoscopic (UCEIS; *p* = 0.0006) *Conclusion:* A relapse is associated independently with younger age, the extension of the disease (E2-E3), increased FC level, C reactive protein, hemoglobin concentration, SCCAI index and UCEIS score.

## 1. Introduction

Ulcerative colitis (UC) represents a chronic disease that causes inflammatory mucosal damage, which affects the patient’s quality of life both through the clinical manifestations of the disease and through invasive monitoring to assess the severity of the disease (endoscopic examination) [1,2,3,4]. The patient often has manifestations that reveal intestinal lesions (abdominal pain, bloody diarrhea, weight loss, anemia) but also extra-digestive manifestations (arthritis, uveitis, skin lesions) [3]. Because the disease is lifelong with recurrent episodes of relapsing and remission, the main goal is to suppress the mucosal inflammation by inducing and sustaining the clinical remission.

Nowadays it is important to acquire mucosal healing objectified by ileo-colonoscopy even if it is expensive, invasive, and with risk for the patients because clinical resolution of symptoms does not mean the absence of mucosal inflammation [1,2,3,4,5,6,7,8]. During recent years, physicians have been interested to find biomarkers that could replace the colonoscopy and monitor patients closely [4,5]. There were reports of alternative tools such as fecal markers (calprotectin or lactoferrin) which are correlated with the endoscopic scores [9,10,11,12,13]. Fecal calprotectin represents a calcium-binding protein derived from colonic mucosa neutrophils [9,10]. Lactoferrin is also secreted from the mucosal colonic neutrophils and during the inflammation is secreted into the stool and is stable for four days [9,10].

Mucosal healing is established during the endoscopy and by collecting samples for the anatomopathologist who can reveal histological remission. Currently, two endoscopic score systems are used in clinical practice, the Mayo Endoscopic Score (MES) and the Ulcerative Colitis Endoscopic Index of Severity (UCEIS) [1,2,3,4,5,6,7,8]. Since 1987, MES is used in clinical trials and defines the mucosal healing [1,2,3,4,5,6,7,8]. The UCEIS score includes the assessment of the vascular pattern (scored 0–3), bleeding (scored 0–3), as well as the presence of erosions and ulcers (scored 0–3). Depending on the UCEIS score obtained, the disease activity could be classified into one of 4 categories, which are clinical remission (UCEIS 0–1), mild (UCEIS 2–4, moderate (UCEIS 5–6) and severe (UCEIS 7–8) [1,2,3,4,5,6,7,8].

The Ulcerative Colitis Endoscopic Index of Severity (UCEIS) accurately reflects the clinical outcomes in patients with UC on induction therapies and also predicts medium-long-term prognosis on those patients [1,2,3,4,5,6,7,8]. In the last 10 years, studies have been published evaluating the association between fecal markers and the UCEIS which have shown that fecal calprotectin could be a reliable parameter of mucosal inflammation that could be correlated with endoscopic and histological grading of disease activity in inflammatory bowel disease [1,2,3,4,5,6,7,8].

The Simple Clinical Activity Index of Ulcerative Colitis (SCCAI) is an index of the severity of the disease that includes six variables: general well-being, number of bowel movements during the day and night, blood in the stool and extracolonic manifestations of ulcerative colitis [14,15,16,17,18]. This index does not require a doctor’s evaluation, laboratory tests or invasive tests (endoscopy) [14,15,16,17,18]. Studies have shown that it is a test of reliability and can define disease remission if the score is <2.5 and is well correlated with invasive scores to define disease activity [14,15,16,17,18].

In this study, our aims were to confirm the correlation between biomarkers and endoscopic disease activity and to define the optimal cut off value to detect clinical and endoscopic remission.

## 2. Methods

### 2.1. Study Design and Patients

This is a prospective study that began in January 2019 and in which all the patients were followed for one year. All patients included in the study were diagnosed with ulcerative colitis in clinical remission established by colonoscopy at the University Emergency Hospital of Bucharest. The inclusion criteria were: patients with UC diagnosed before June 2018, age ≥ 18 years and in clinical remission (over 6 months). The exclusion criteria were: active UC, pregnancy, unclassified inflammatory bowel disease, pouchitis or colectomized patients.

Clinical remission was assessed using Simple Clinical Colitis Activity Index (SCCAI < 2.5) and endoscopic remission that was defined using Ulcerative Colitis Endoscopic Index of Severity (UCEIS 0–1). Follow-up was done every 3 months by making SCCAI, fecal calprotectin, C-reactive protein and hemoglobin. All the patients included in the study performed colonoscopy or flexible rectosigmoidoscopy at 0, 6 and 12 months. Two independent IBD physicians retrospectively, separately and blinded reviewed images from the endoscopic reports and graded the endoscopic activity. The highest score was chosen as the overall score after resolving the discrepancies between the two endoscopists.

### 2.2. Statistical Analysis

The statistical analysis was done using the STATA 13/MP statistical software. Sensitivity and specificity with 95% CIs for finding UCEIS levels was established according to calprotectin, C-reactive protein, and hemoglobin results. We analyzed the receiver operating characteristic curve and the area under the curve (AUC) to evaluate the appropriate cut-off values for calprotectin, C-reactive protein, and hemoglobin. Data were analyzed for normality (using the Kolmogorov-Smirnovff test and the Shapiro–Wilk test); consequently, parametric/non-parametric tests were used. The threshold of statistical significance was *p* ≤ 0.05.

### 2.3. Ethical Considerations

All patients had written informed consent for inclusion before enrolling in the study. The study was conducted in accordance with the Declaration of Helsinki, and the protocol was approved by the Local Ethics Committee of University Emergency Hospital of Bucharest (37322/30.07.2020, 16 November 2020).

## 3. Results

### 3.1. Baseline Characteristics of the Patients

The study included 59 patients with age between 21 and 77 years. The median age was 47 years with a range spanning from 21 to 77 years old. Of those, 55.9% (33 patients) were male and 12%were smokers. Montreal classification of extent of UC was as follows: ulcerative proctitis (E1)–18.6%, left sided UC (E2)–40.7% and extensive UC (E3)–40.7%. At the baseline 98.3% of the patients were taking oral 5-aminosalicylates, 44.1% were receiving azathioprine and 49.2% anti-TNF. The two anti-TNF molecules used on this patients were infliximab (32.2%) and adalimumab (17%). At onset, the median period of remission of the disease was 30 months. During the follow-up, 21 patients (35.6%) had a relapse between 3 and 12 months. Baseline characteristics of the patients are summarized in Table 1.

### 3.2. Fecal Calprotectin

#### 3.2.1. Fecal Calprotectin Measurement

The measurement of calprotectin was done in an external associate laboratory for all included patients, using quantitative enzyme-linked immunosorbent assay (ELISA). All determination were blinded by the current clinical and endoscopic disease activity. The results were expressed as µg/g and the lower and the upper limits of detection were 10 and 4000 µg/g, respectively.

#### 3.2.2. Relationship between Fecal Calprotectin and Endoscopic Activity

At the baseline the median value of fecal calprotectin was 47 µg/g and with a of range 6.9–188 µg/g. During the follow-up, the fecal calprotectin (FC) measurement was significantly increased at 3 months (median value, IQR µg/g; 745, 14–4000) and at 6 months (median, IQR µg/g; 650, 4.5–3000) compared to baseline (*p* ≤ 0.05). FC level was associated with a mild increased risk of relapsing (Table 2), corresponding to an odds ratio (OR) of 1.05 (95% CI, 1.03–1.06; *p* = 0.0001). Additionally, fecal calprotectin can predict the relapse of the patient’s disease in our study group, by obtaining an area under the ROC curve of 0.9260 (*p* = 0.0005) (Figure 1).

According to the ROC model we calculated the correctly classified cases and the cut-off for the optimal correctly classified values for fecal calprotectin concentration during the baseline until the end of study (sensitivity, specificity, positive predictive and negative predictive value). The right cut-off for correctly classifying patients as disease remission/disease activity using calprotectin value (i.e., under this value the patients were in clinical remission and above this value they have disease activity) was 88 µg/g with a sensitivity of 72.38%, specificity of 95.68%, positive predictive value of 86.36% and a negative predictive value of 85.99%.

### 3.3. C-Reactive Protein

C-reactive protein, another marker of inflammation that can be used in determining the activity of inflammatory bowel disease, was assessed in the study, as well as its association with disease relapse. The measurement of CRP revealed higher values at 3 months (median, range, mg/dL; 1.8, 0.14–58.9), at 6 months (median, range, mg/dL; 2.36, 0.12–45.8) and at 9 months (median, range, mg/dL; 2, 0.12–25.9) compared to baseline (median, range, mg/dL; 1.26, 0.12–6.98).

The analysis of variance of CRP showed a value of *p* = 0.01. This biomarker was associated also with an increased risk of relapse (Table 2) with an OR of1.74(1.51–2.00; *p* ≤ 0.0001). During the analysis of the data we obtained a cut-off point of CRP value (5.5 mg/dL) (sensitivity 63.81%, specificity 93.16%, positive predictive value 83.75%, negative predictive value of 82.33%) and an area under the ROC curve of 0.8699 (Figure 2).

### 3.4. Haemoglobin

All patients were monitored during the study by hemoglobin value, biomarker which plays an important role in predicting a relapse (*p* = 0.0393). The value of hemoglobin was lower on those who presented a relapse at 3 months (median, range, g/dL; 12.9, 9.2–15.2) and 6 months (median, range, g/dL; 12.5, 9.5–15.3) compared to baseline (median, range g/dL; 12.8, 11–16.1). During the follow-up the area under the ROC curve was 0.7538 (Figure 3). The best cut-off point of hemoglobin values was 11.9 g/dL (sensibility 50.48%, specificity 96.84%, positive predictive value 89.83%, negative predictive value 80.34%) (Figure 3).

### 3.5. Simple Colitis Clinical Activity Index (SCCAI)

Clinical assessment of disease activity was done using the Simple Colitis Clinical Activity index (SCCAI) at every visit during the study. Association between SCCAI and relapse revealed an OR of 14.04 (95% CI, 6.88–28.66, *p* ≤ 0.00001) (Table 2). For this parameter the area under ROC curve was 0.8510 (Figure 4). For SCCAI score de cut-off point was 1 calculated by the ROC model with a sensitivity of 73.33%, specificity of 94.21%, positive predictive value of 87.50% and negative predictive value of 86.47% (Figure 4).

### 3.6. Ulcerative Colitis Endoscopic Index

The assessment of mucosal healing was done according to the Ulcerative Colitis Endoscopic Index (UCEIS score) at the baseline, 6 and 12 months. At the 6 months the mean value of UCEIS was 1.3 with an IQR 0–7 and at the 12 months a mean of 0.6 and an IQR of 0–4 (*p* ≤ 0.0006). The OR for UCEIS score was 34.50 (95% CI, 12.06–98.74; *p* ≤ 0.00001). The area under ROC curve to predict UC relapse using UCEIS score was 0.8906 (Figure 5).

Survival curves (with relapse as event) are depicted in the figures below; all log-rank tests yielded significant results (*p* < 0.0001) (Figure 6, Figure 7, Figure 8, Figure 9 and Figure 10.)

On multivariate Cox proportional hazard analysis, a relapse was associated independently with the extension of the disease (E2-E3), increased FC level, C reactive protein, hemoglobin concentration, SCCAI index and UCEIS score (Table 3).≥

## 4. Discussion

The recurrence of inflammatory bowel disease is mostly unpredictable. It is important for clinicians to identify those patients at higher risk of imminent clinical relapse of disease at a presymptomatic stage. Over the time, a marker, fecal calprotectin, has been found to be more sensitive than endoscopy to closely correlate with histological activity [1,2,3,4,5,6,9,10,11,12,13,18,19]. Studies have shown that the value of calprotectin may better reflect the activity of the disease in ulcerative colitis than in Crohn’s disease [20]. Another representative inflammatory biomarker is CRP which is used in monitoring disease activity.

In 2017, Lee SH and colleagues evaluated the correlation of the FC levels with two endoscopic severity indices (MES and UCEIS) [7].

The aim of our study was to define those reliable biomarkers that are correlated with clinical and endoscopic scores to monitor UC activity and to provide patients with disease control, but also to prevent unnecessary examinations. Patients with inflammatory bowel disease in our clinic made us develop a study similar to those in the literature, because years of illness, frequent recurrences, repeated endoscopic examinations make them fear and become unfulfilled in the treatment and monitoring plan. Another aspect is that Romanian patients do not always have the necessary financial support to be evaluated with fecal biomarkers (fecal calprotectin) whenever they have increased bowel movement without a clinical visit, which sometimes they cannot afford. Another objective of the study was to set the cut-off value of these parameters to establish clinical and endoscopic remission. A secondary goal that would require the attention of any clinician is to find those markers that can predict a future episode of relapse, but also the time to relapse.

Our study shows that 35.6% of patients had a relapse during the study. Disease progression was associated with elevated fecal calprotectin, C-reactive protein and UCEIS score) during the 12 months of the study. As in other studies, serum and fecal inflammatory biomarkers (CRP, fecal calprotectin) were also compared with endoscopic appearance to monitor disease activity [4,5,6]. In our study, the association of FC with UCEIS was not significant for disease relapsing compared to CRP which had a better correlation with UCEIS. The two endoscopic experts used the UCEIS endoscopic score to establish the presence of disease activity, without knowing the patient’s biological parameters.

Several studies have shown that fecal markers such as calprotectin and hemoglobin concentration could also be correlated with MES [5,6,7]. In our study, patients with low hemoglobin values associated a new relapse. Regarding to the endoscopic activity scores, UCEIS, shows a significant correlation with levels of FC, according to the observations of previous studies [1,2,3,4,5,6,7,8]. Another parameter followed in our study is SCCAI, which correlates with UCEIS and FC, CRP and hemoglobin values. In 2018, MShoichiro and colleagues demonstrated that fecal markers are the most sensitive tools for detecting macroscopic inflammation and are correlated with UCEIS [3]. Following this study, we observed that for our group of patients the cut-off values of fecal calprotectin, CRP and hemoglobin were 88 µg/g, 5.5 mg/dL and 11.9 g/dL, respectively.

We observed in our study that the relapse of the disease is associated with the patient’s age, but also with the extent of the disease (E2-E3), these predictive parameters of the recurrence of ulcerative colitis were highlighted in other studies [13].The limitation of this study is related to the small number of patients, the relatively short study period (12 months) and that it was conducted in a single center that can reduce heterogeneity. Clinicians have tried to find clues that they have validated or need to be validated by the best statistical criteria. These will help to establish the therapeutic behavior, but will also encourage doctors to make efforts in appropriate treatment goals, leading to the improvement of the evolution of patients with IBD, but also of their quality of life.

In conclusion, we showed in this study that a relapse was independently associated with the extension of the disease (E2-E3), increased FC level, C reactive protein, hemoglobin concentration, SCCAI index and UCEIS score.

## Figures and Tables

**Figure 1 medicina-57-00031-f001:**
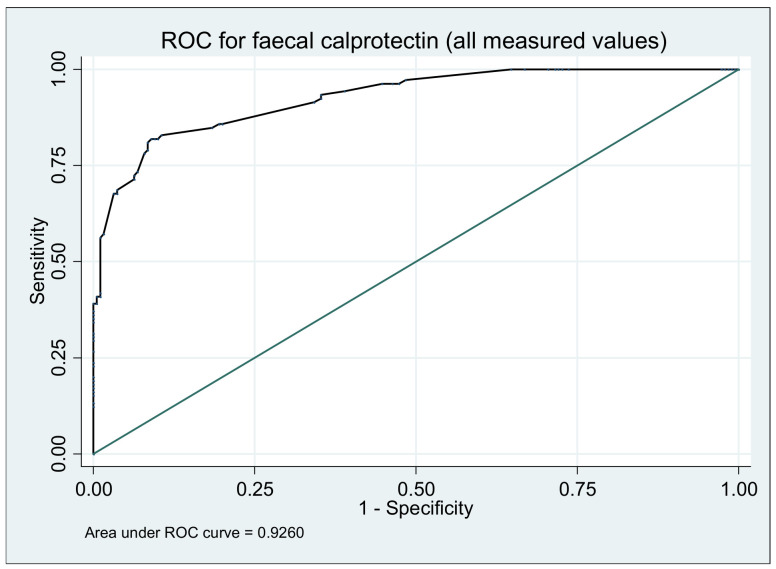
Area under the receiver operating characteristic (ROC) curve to predict ulcerative colitis relapse using fecal calprotectin measurement during 12 months.

**Figure 2 medicina-57-00031-f002:**
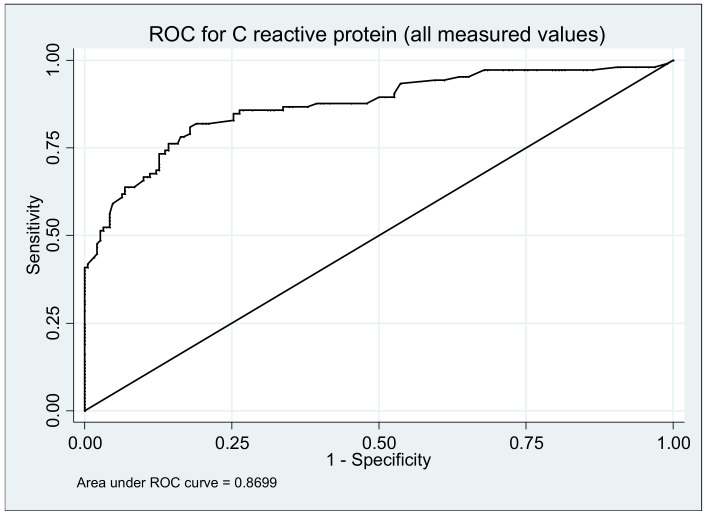
Area under the receiver operating characteristic (ROC) curve to predict ulcerative colitis relapse using C reactive protein determination during 12 months.

**Figure 3 medicina-57-00031-f003:**
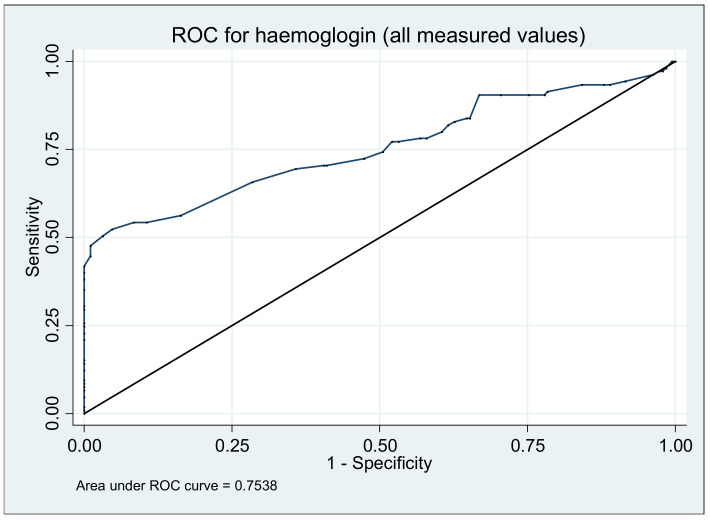
Area under the receiver operating characteristic (ROC) curve to predict ulcerative colitis relapse using hemoglobin concentration measurement during 12 months.

**Figure 4 medicina-57-00031-f004:**
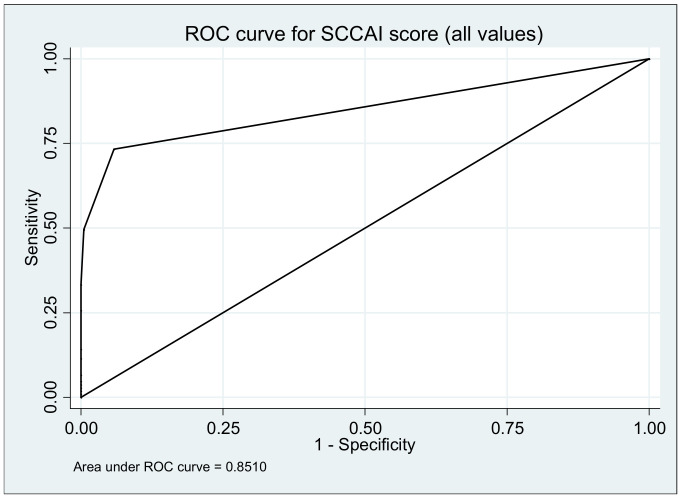
Area under the ROC curve to predict ulcerative colitis relapse using the Simple Clinical Activity Index of Ulcerative Colitis (SCCAI) score during 12 months.

**Figure 5 medicina-57-00031-f005:**
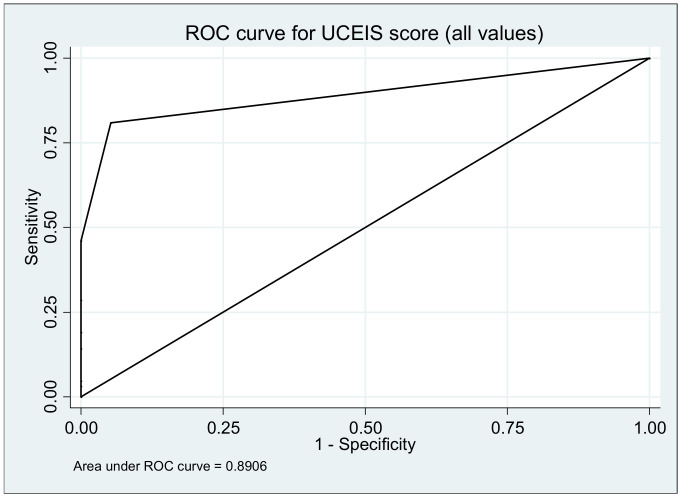
Area under the receiver operating characteristic (ROC) curve to predict ulcerative colitis relapse using the Colitis Endoscopic Index of Severity (UCEIS) score during 12 months.

**Figure 6 medicina-57-00031-f006:**
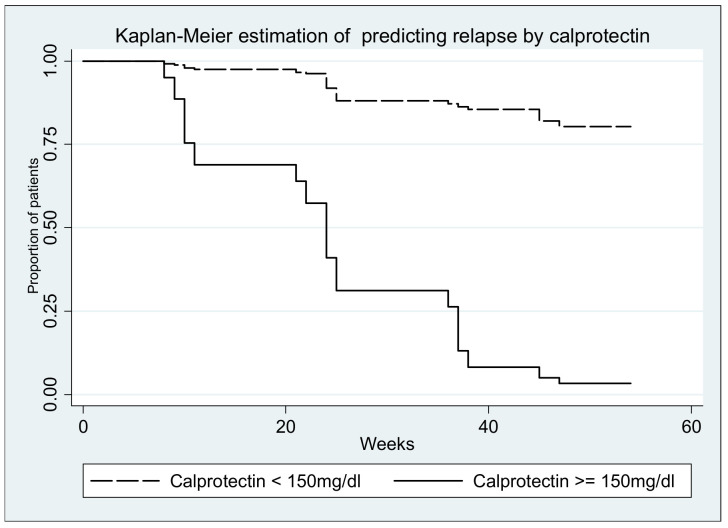
Log-rank test for equality of survivor functions (calprotectin) *p* < 0.00001.

**Figure 7 medicina-57-00031-f007:**
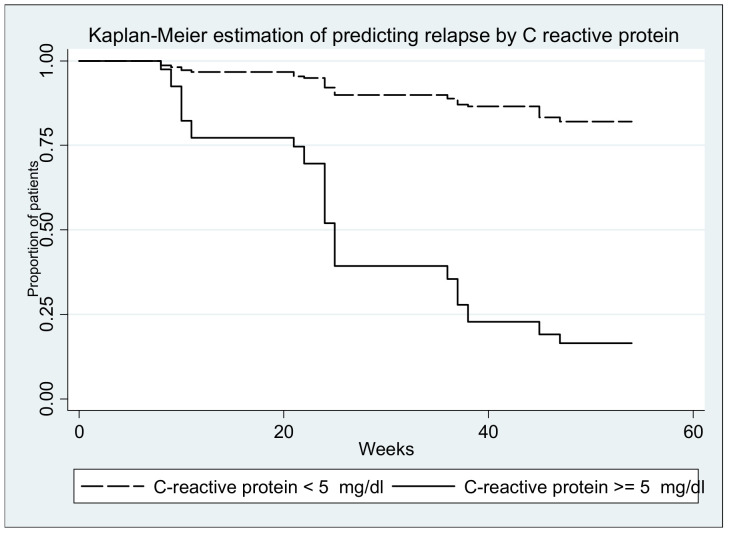
Log-rank test for equality of survivor functions (C reactive protein) *p* < 0.00001.

**Figure 8 medicina-57-00031-f008:**
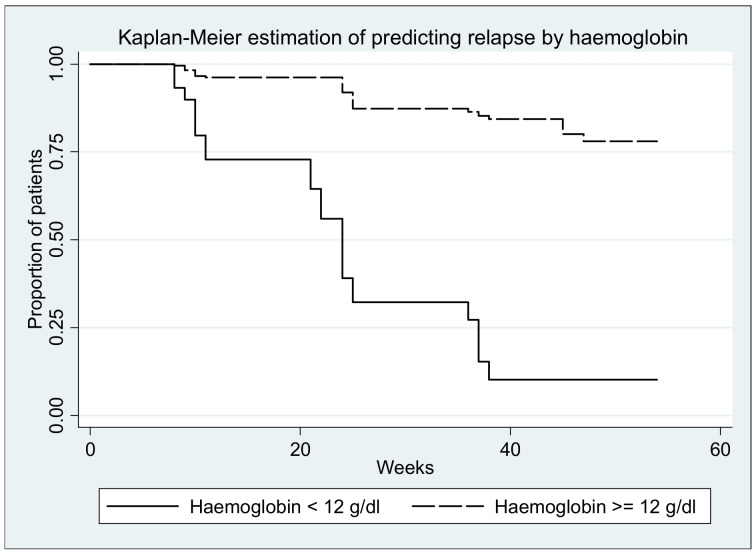
Log-rank test for equality of survivor functions (hemoglobin) *p* < 0.00001.

**Figure 9 medicina-57-00031-f009:**
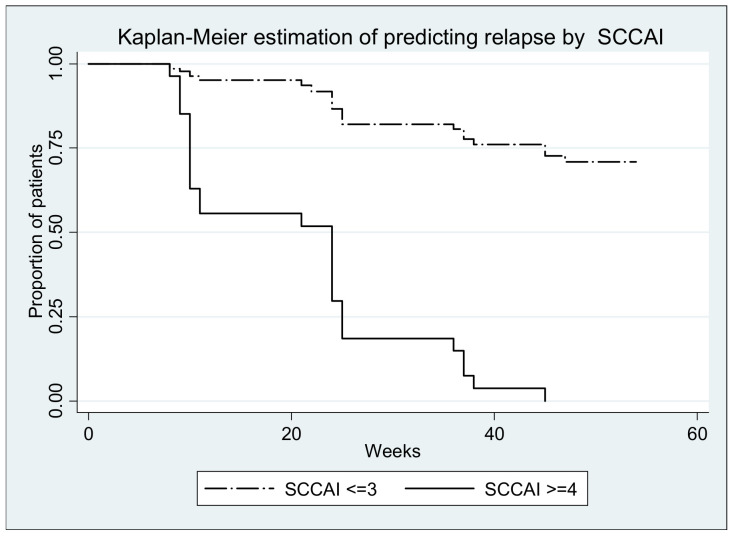
Log-rank test for equality of survivor functions (SCCAI) *p* < 0.00001.

**Figure 10 medicina-57-00031-f010:**
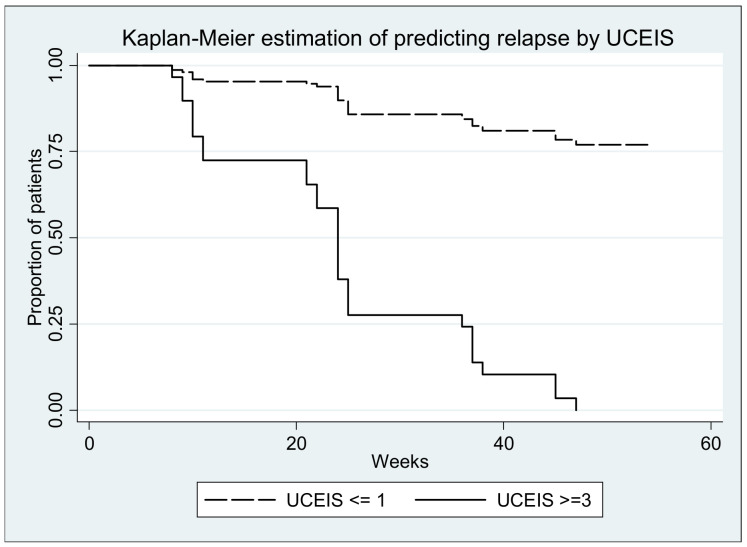
Log-rank test for equality of survivor functions (UCEIS) *p* < 0.00001.

**Table 1 medicina-57-00031-t001:** Socio-demographic and clinical characteristics of patients at the baseline.

Age (Years, Median, Range)	47 (21–77)
Gender (% male)	55.9
Smoking (% smokers)	11.9
Montreal classification of extent of UC (%)	
E1	18.6
E2	40.7
E3	40.7
Concomitant medication at baseline %	
5-ASA (Mesalazine)	98.3
Azathioprine	44.1
Corticosteroids	10.2
Infliximab	32.2
Adalimumab	17.0
Time of remission of disease until inclusion in the study (months; median range)	30 (4–84)

SD, standard deviation; UC, ulcerative colitis, 5-ASA, 5-aminosalicylates.

**Table 2 medicina-57-00031-t002:** Association between different parameters (odds ratio, OR) and relapse and the receiver operating characteristic (ROC) curve predicting the relapse over the time.

Parameters	OR (95% CI)	*p* (Z Test)	ROC Area (95% CI)	*p* (Chi^2^)
FC	1.05 (1.03–1.06)	≤0.0001	0.93 (0.90–0.96)	0.0005
CRP	1.74 (1.51–2.00)	≤0.0001	0.87 (0.84–1.00)	0.0047
HB	2.30 (1.81–2.93)	≤0.0001	0.75 (0.69–0.82)	≤0.0001
SCCAI score	14.04 (6.88–28.66)	≤0.00001	0.85 (0.81–0.90)	≤0.0001
UCEIS score	34.50 (12.06–98.74)	≤0.00001	0.89 (0.84–0.94)	0.11

**Table 3 medicina-57-00031-t003:** Hazard ratio (HR) of parameters associated with relapse (*n* = 59).

Parameters	HR (95% CI)	*p* (Z Test)
Gender (Women vs. Male)	0.81 (0.54–1.19)	0.281
Age (over 60 yrs vs. under 60 yrs)	1.01 (1.00–1.02)	0.045
Extent of ulcerative colitis (E2-E3 vs. E1)	2.31 (1.69–3.14)	≤0.0001
Smoking (Smoker& ex-smoker vs. Non-smoker)	1.10 (0.94–1.28)	0.224
Fecal Calprotectin (0–12 months)	1.001 (1.0010–1.0015)	≤0.0001
C reactive protein (0–12 months)	1.10 (1.08–1.12)	≤0.0001
Hb concentration (0–12 months)	1.39 (1.32–1.47)	≤0.0001
Simple Clinical Colitis Activity Index (SCCAI) over time (0–12 months	1.39 (1.32–1.47)	≤0.0001
Ulcerative Colitis Endoscopic Index (UCEIS) over time (0–12 months)	1.70 (1.53–1.90)	≤0.0001

## Data Availability

Data is contained within the article.

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
