# Peer review of "Correlation of Biomarkers with Endoscopic Score: Ulcerative Colitis Endoscopic Index of Severity (UCEIS) in Patients with Ulcerative Colitis in Remission"

_medicina, 2020, doi:10.3390/medicina57010031_

Round 1
Reviewer 1 Report
Review of: Correlation of biomarkers with endoscopic score - Ulcerative Colitis Endoscopic Index of Severity (UCEIS) in patients with ulcerative colitis in remission
General: In this study, the authors prospectively included patients with UC in remission and subsequent examined the risk of relapse and associated this with clinical symptom scores and biomarkers of inflammation. While this is always interesting, it has been done in several studies previously. The authors should recognize this and include a more thorough literature revied in the introduction.
Unfortunately, this paper is quite difficult to read due to two main factors. 1: The written English in this paper is suboptimal and needs to be revised throughout the paper. I have not gone into this issue in my comments below. 2: It is difficult to see what the main findings in this paper are. These should relate strongly to the aim and should be presented as the first thing in the discussion.
Abstract: The abstract should present the most important finding of a study. Is it the opinion of the authors that this is the case in the submitted abstract. Also, all data that leads to the drawn conclusions should be presented. The abstract needs to be revised.
The first sentence is ambiguous. As it is now, it implies that the diagnosis per se is unfavorable. While this is of course right, I think the authors mean to say that the prognosis relates to endoscopic and histologic findings? The second sentence need to be revised. The authors state, that males are more often affected than females. The data does not allow such a statement. In results, the authors state that FC (remember to define at first use and use this throughout the paper) is “significantly higher”. Higher than what? Also, I think increased is the correct term to describe this. For CRP it is stated the it is “high” at 3 and 6 months. Is it not compared to something? Stating a p value as 0.0121 as a bit excessive. 0.01 Is sufficient. A HR for SCCAI at 4 is presented. What is this a risk of? I do not understand the sentence presenting the UCEIS results.
Introduction: The us of bullits in the introduction is atypical. The authors state that “a few” studies have been published correlating FC and endoscopic findings and present 4 references. I think the literature on this issue is quite extensive and further details could have been included on this issue. It would be helpful to the reader if all biomarkers used in this study were presented in the introduction.
Methods. The authors state that they included patients who had been in remission for 6-12 months. Did they not include patients who had been in remission for longer than 12 months? The authors excluded patients who had pouchitis. Did they include colectomised patients?
Statistics: Data not following normal distribution should be presented using non-parametric analyses. Hence, age (only patients >18 are included, so cant be normally distributed) and FC (measurements are capped at 4000 so cant be normally distributed) should be presented by medians instead. This is also the case for time since diagnosis and it is underlined by the stated median time of 2.6 +/- 5.6 years, which objectively does not make sense.
The survival analysis used is not presented here and it would be helpful to understand the results.
Results: As I understand, all patients included were in remission. How did the authors categorize patients into the Montreal classification? Was this from previous endoscopies? A section on this should be added to methods.
It is stated that cox proportional hazard is used. This analysis is usually presented by hazard rationes. However, in Table 2 Odds ratios are presented. Can the authors elaborate on this?
It is unclear to me what exactly the survival analysis included. How was event and strata defined.
In the ROC analysis, it is unclear to me, what variable was used to define remission. UCEIS? SCCAI? Other?
Regarding hemoglobin, it is stated that the levels at 3 and 6 months were lower in patients with relapses. Data and statistical analysis showing this should be presented.
Tables:
The setup of the tables is confusing. Especially Table 1. Please align the first column to the right.
Table 3: I think the statistics should be revised for this table. I don’t understand how the FC HR of 1.001 with a 95%CI of 1.001-1.0015 can give a p value of 0.0001. Also, it is strange that 6 of the 9 tests show a p value of exactly 0.0001. Might it be <0.0001?
Discussion: It would be helpful to the reader if the authors started this section with the main findings of the study. In general, I think the findings of this study needs to be discussed in further detail.
The authors state that a number of variables are associated to increased disease progression. How is increased disease progression defined? In line 195, I don’t understand the sentence starting with “both FC and CRP … “.
Lactoferrin is mention both in the introduction and in the discussion. Maybe the authors should focus on the biomarkers measured in the study.
All p values should be presented in the results section, not in the discussion.
In line 205 the authors state that fecal markers are highly sensitive. This could be specified.
It is stated that the risk of relapse is associated with age. I am unsure whether a HR of 1.01 (CI 1.0-1.02) justifies this statement. This clinical significance of this finding should be discussed.
The authors conclude that SCCAI and UCEIS scores are associated with risk of relapse. Wasn’t these scores how patients in remission were defined?
Author Response
Thank you for giving us the opportunity to submit a revised draft of our manuscript.
We appreciate the time and effort that you have dedicated to providing your valuable feedback on the manuscript. We are grateful to the reviewers for your insightful comments on the paper. We have highlighted the changes within the manuscript.
Here is a point-by-point response to your comments and concerns.
Sincerely,
MD Petruta Filip

Reviewer 2 Report
This is a prospective study focusing on one of the hot topics in IBD field: prediction of disease relapse. Apart from the need of extensive english editing, I have some concerns:
- in the abstract (line 30) they cite a 9 months follow up, but the study schedule is 0, 6 and 12 months. what is this 9 months follow up(maybe it was a miswritten number?)?
- In line 112 they say that table 1 shows basal charateristics of both patients who were in remission and who relapsed during follow up, but table 1 reports the basal charateristics of the whole population (not divided in patients who mantained remission and those who did not).
- I don't understand why in line 119-121 they say that 150 mcg/kg was considered as an initial threshold for disease activity. Was it an inclusion criteria? (in this case it should be stateds in the methods' section).Were the patients enrolled and then had fecal calprotectin dosed and subsequently defined as in remission on not in remission at time 0? why did they define a threshold if one of the endpoints of teh study was to find one?
- In results section they report in the text HR for each variable, while in table 2 they report OR. I think they should report the same parameter in both.
- both in monovariate analysis and in multivariate analysis I can see that fecal protectin is one of the less strongly associated with relapse as the HR/OR is very close to 1. The major predictive variables seem to be disease extent, HB level, clinical and endoscopical scores both in terms of HR and in terms of OR and both in monovariate and in multivariate analysis. I think that authors should explain it better in the discussion
- in discussion section (line 193) they say that endpoint was reached by 64,4% of patients but the endpoint was not explicitated in the methods section and this data was not reported in the results section.
- I was wondering if the authors assessed therapeutical differences between the patients who relapsed and those who did not during the follow up? did patients who were undergoing anti TNF or immunosuppressant therapies relapse at the same rate of the patients undergoing mesalamine only?
- How long after elevation of fecal calprotectin, low HB levels, ecc did the patients experience relapsing? is it possible that these variables indicate an ongoing relapse and do not predict a future one?
Author Response
Thank you for giving us the opportunity to submit a revised draft of our manuscript.
We appreciate the time and effort that you have dedicated to providing your valuable feedback on the manuscript. We are grateful to the reviewers for your insightful comments on the paper. We have highlighted the changes within the manuscript.
Here is a point-by-point response to your comments and concerns.
Sincerely,
MD Petruta Filip

This manuscript is a resubmission of an earlier submission. The following is a list of the peer review reports and author responses from that submission.
Round 1
Reviewer 1 Report
Thank you for this interesting article. You have shown some correlation between faecal calprotectin, CRP and haemoglobin levels in predicting UC relapse as confirmed by endoscopy. From my point of view it would be helpful if you could clarify a few points:
Line 106 - ethical considerations. You need to include what Ethics/regulatory approval you gained for the study.
Line 124 - You mention that the cut-off for FC was set at 150 but how/why did you choose that cut off and what did it mean? for example did anything below 150 indicate remission or absence of inflammation etc.
Line 139 - You mention that the right cut off for FC was 88 but again what do you mean by 'right', what did that cut off mean or indicate?
Line 162 - You mention the best cut off for haemoglobin - what did the cut off indicate and was it the same for men and women (who often have different cut off levels for normal haemoglobin levels)
UCEIS - I am not familiar with the scores or staging in UCEIS, it would be helpful for the reader if you could give a brief description of this for example is a lower score no inflammation, higher score worse inflammation. It would help with understanding the results.
Endpoint - you mention only 64% of respondents reached the endpoint - do you mean only 64% completed the study to the final follow up at 12 months? If so could you indicate that more clearly. It would be helpful to indicate how many dropped out and the main reasons why if you have that data.
Thank you, I look forward to reading the revised version
Author Response
On behalf of all the authors,
We would like to thank you for the comments on our study, which were beneficial for us.
Best regards,
All team

Reviewer 2 Report
This is a prospective study aiming to confirm the correlation between biomarkers and endoscopic disease activity and defining an optimal cut off value to detect clinical, endoscopic, and histological remission in UC patients. However, as also indicated in the introduction, similar studies have been done before indicating fecal calprotectin as a marker for monitoring IBD (remission and relapse) and there are current clinical guidelines indicating its widespread application. It would be necessary for the authors to clear the novelty the current study seeks to present throughout the manuscript and/or include the necessary edits to do so.
I have noticed several issues with methods and data reports that must be addressed:
-
- p1 Line 34 (abstract)/ Line 128 (Results): p=0.0536 was considered statistically significant. However, methods indicate that the threshold of statistical significance was considered P ≤ 0.05.
- Table 1 may represent patients as the different categories they fall into. Pulling altogether might lead to a misunderstanding of the data. Also, the units for time, etc. should be included.
- Table 2, ROC area CAN NOT be over the 1 in value. Please reconsider data analysis and reporting. the last column could be either removed, or a description is needed as to why respective values seem to be blank and should be reported.
- Small pvalues may be reported as less the 0.01 or 0.001.
- Methods to determine cutoffs from ROC curves should be included. e.g. Youden's J statistic
- the rationale for the following should be described: The highest score was chosen as the overall score after resolving the discrepancies between the two endoscopists.
Author Response
Dear reviewer, Thank you for your comments which were well made and we appreciate this as theycould lead to a real disability on our work.
We worked hard to carry out this study and it would have been a shame for
our inattention to process statistical data in the text to lead to the
non-publication of this paper.
Thank you again!
Best regards,
All team

Reviewer 3 Report
Dr. Pop and colleagues investigated the role of biomarkers and endoscopic disease activity in predicting ulcerative colitis relapses in a cohort of patients in remission. Moreover, they aimed at defining optimal cut off values of some biomarkers in order to detect clinical, endoscopic and histological remission or relapse.
The study is interesting and the topic is current. However there are some points that need attention and clarification.
- abstract: should be rewritten at least in the result section (it is not clear); actually the authors did not assess histologic activity, so this statement has to be removed;
- introduction: the aims of the study should be clearly presented at the end of the introduction;
- methods: it is not clear to me if the study is prospective as stated in row 77, or retrospective, as suggested by lines 87-90: why was the endoscopic evaluation performed retrospectively?
- results: baseline characteristics of patients are repeated in the text and in table 1; moreover there are some inconsistencies (i.e. smokers were 7% or 11.9%? Time of remission higher than time since diagnosis?); Lines 141-142: the sentence makes no sense and should be rephrased; line 146: hazard ratio of: missing value;
- it is not clear to me from table 3 and lines 189-191 if younger or older age and higher or lower Hb levels were associated with relapse; please clarify.
- Line 222: the fact that it was a single center study can reduce heterogeneity (in my opinion this is a pro), but reduces replicability and generalizability of the study findings.
Author Response
On behalf of the entire team,Thank you for your complete review.
Your comments have led to an improvement in the quality of this article.
Best regards,
All team

Round 2
Reviewer 2 Report
The authors have referred to some of the concerns however the following are necessary:
1- The authors have not added any rationale for the necessity to re-evaluate the already confirmed correlation of IBD activity/remission to Calprotectin.
2- The authors just considered a p-value of 0.0536 equal to p≤0.05 without any new data which I don't consider correct. Also, the authors have not explained why the consider p= 0.05 statistically significant.
3- Applications are tools for statisticians to use a different variety of tests and methods. I did not see the reply to my comment as adequate in this regard. Please specify methods used to determine cutoffs from ROC curves.
4- Table 2, the column for p(chi2) is blank and the authors have not addressed my question regarding the necessity of this column.
5- Although the authors have replied "Thank you for your attention!Small pvalues may be reported as less the 0.01 or 0.001. " there are discrepancies throughout the manuscript ( Line 151, Table 2, line 172). please keep reporting consistent.
6- Re the following " Two independent IBD physicians retrospectively, separately and blinded reviewed images from the endoscopic reports and graded the endoscopic activity. The highest score was chosen as the overall score after resolving the discrepancies between the two endoscopists." I have already asked the rational, but the authors replied as"discrepancies between the endoscopists were not very large, they varied in 10 cases. In all cases, the highest score values were chosen."
This might be a selection bios, please include data and/or add further details on why the highest score was selected, and what was the difference between scores? were they significant?
7- Regarding data/table 1, different extents of the disease (E1 to E3) may have different presentation re biomarkers as well. If the data was described and matched in different categories it would have been better.
Reviewer 3 Report
The authors have addressed reviewers' comments.